# Stabilizing Li-Rich Layered Cathode Materials Using a LiCoMnO_4_ Spinel Nanolayer for Li-Ion Batteries

**DOI:** 10.3390/nano12193425

**Published:** 2022-09-29

**Authors:** Hsiu-Fen Lin, Si-Ting Cheng, De-Zhen Chen, Nian-Ying Wu, Zong-Xiao Jiang, Chun-Ting Chang

**Affiliations:** Department of Materials Science and Engineering, National Formosa University, Yunlin 632301, Taiwan

**Keywords:** layered cathode, nanolayer spinel coating, cycling ability, rate capability

## Abstract

Lithium–rich cathodes have excess lithium in the transition metal layer and exhibit an extremely high specific capacity and good energy density. However, they still have some disadvantages. Here, we propose LiCoMnO_4_, a new nanolayer coating material with a spinel structure, to modify the surface of lithium cathode oxide (Li_7/6_Mn_1/2_Ni_1/6_Co_1/6_O_2_) with a layered structure. The designed cathode with nanolayer spinel coating delivers an excellent reversible capacity, outstanding rate capability, and superior cycling ability whilst exhibiting discharge capacities of 300, 275, 220, and 166 mAh g^−1^ at rates of 0.1 C at 2.0−4.8 V formation and 0.1, 1, and 5 C, respectively, between 2.0 and 4.6 V. The cycling ability and voltage fading at a high operational voltage of 4.9 V were also investigated, with results showing that the nanolayer spinel coating can depress the surface of the lithium cathode oxide layer, leading to phase transformation that enhances the electrochemical performance.

## 1. Introduction

With the intensification of climate change and environmental pollution, lithium–ion batteries (LIBs) are being increasingly pursued to power electric vehicles (EVs) and store renewable energy [1,2,3]. A safe, long cycle life with higher energy density and good chemical performance is desired for LIBs. Increasing the capacity of the cathode and the charging voltage of the cells are potential methods for achieving this. A lithium–rich cathode with excess lithium in the transition metal layer shows an extraordinarily high specific capacity after an irreversible oxygen release activation process at approximately 4.5 V, which may be a promising solution for producing next–generation cathode materials. However, owing to the shortcomings of the low initial Coulombic efficiency, poor rate capability, and voltage attenuation upon cycling, lithium–rich cathodes are difficult to commercialize [4,5,6,7].

Recently, the disadvantages of lithium–rich cathodes were found to be associated with the poor mass transport of the rearranged surface once the activation of Li_2_MnO_3_ exceeds 4.5 V [8]. In addition, the elimination of the oxygen atom and Li^+^ vacancies from the layered lattice, which decreases the number of sites for insertion and extraction of Li^+^ in subsequent cycles, is the reason for the irreversible capacity [9]. These materials also possess fragile surface properties at high potentials, can experience electrolytic erosion, and may induce the dissolution of transition metal ions [10,11,12]. Recently, it was reported that surface modification and bulk doping could potentially overcome these issues.

The surface modification of cathode materials by metal oxide layers such as Al_2_O_3_, NbO_5_, Ta_2_O_5_, ZrO_2_, and ZnO [13,14,15,16,17,18] can decrease side reactions at the electrode–electrolyte interface, but this usually has a negative impact on the rate capability due to the slow Li^+^ or electron–conductivity of the coating layer [19,20]. Recent studies have suggested that coating cathode materials with a layer of electron conductors or Li^+^ conductors, such as carbon [21], AlF_3_ [22], LiNiPO_4_ [23], Li_2_ZrO_3_ [24], or LiMgPO_4_ [25], can enhance the rate capability to some extent, as these can promote charge transfer or Li^+^ diffusion from the surface of the electrode material.

Spinel–type LiNi_x_Mn_2-x_O_4_ (0 < x < 0.5) oxides exhibit great potential as high–power cathode materials for LIBs [17,26,27] because of their high Li^+^ ionic and electronic conductivities [28], high voltage, low cost, and low toxicity. Presently, spinel oxides are a considerable choice to encapsulate Ni–rich layered oxides for identical cubic close–packed arrays [29,30]. Yan et al. [31] designed a composition of a Ni–rich layered core and a LiNi_0.5_Mn_1.5_O_4_–like spinel shell. The heterostructure, with a low degree of Ni^2+^ and Li^+^ disordering, can afford a stable structure framework and high–efficiency paths for the diffusion of lithium ions. Min et al. [32] proposed novel particles, the spinel interphase of which is coherent with the surface of the LiNi_0.9_1Co_0.06_Mn_0.03_O_2_ (NCM) cathode. The spinel–coated NCM structures outperformed the bare sample in terms of the capacity retention rate and rate capability: they conserved more than 90% of the initial capacity at 1C for up to 50 cycles. After cycling, the crack generation was significantly reduced in the secondary particles of the coated structures compared to that of the bare sample.

In addition, some studies have combined the advantages of lithium–rich cathodes and spinel materials to enhance electrochemical performance in terms of kinetics, energetics, and core material stability [33,34,35]. In a study by Choi et al. [36], Li–rich composite oxides were synthesized by doping LiMn_2_O_4_, LiNi_0.5_Mn_1.5_O_4_, and LiCo_0.5_Mn_1.5_O_4_. Of the various composites created, the LiNi_0.5_Mn_1.5_O_4_–embedded sample showed lower formation energy as determined by first–principles calculation [37,38]. Furthermore, the Li–rich oxide composite formed by doping the optimal spinel, LiNi_0.5_Mn_1.5_O_4_, exhibited the best cycling performance with little voltage decay when used in a cylindrical 18650–type full cell. Wu et al. [39] developed a lithium–rich cathode (Li_1.2_Mn_0.6_Ni_0.2_O_2_) coated with a 4 V Li_1+x_Mn_2_O_4_ spinel membrane for high–energy and high–power Li–ion batteries. The modified cathode displayed significant enhancements in specific capacity and cycling ability compared to an unmodified cathode. It yielded a maximal capacity of 295.6 mAh g^−^^1^, and a capacity of 280.0 mAh g^−^^1^ by the 50th cycle, presenting excellent capacity retention at 94.7%. Yu et al. [40] reported an effective approach for synthesizing layered spinel–capped, nanotube–assembled 3D Li–rich hierarchitectures. The resulting material’s unique 3D hollow hierarchical structure significantly shortened electron and ion transfer pathways whilst maintaining reliable structural stability. As a result, the LiMn_1.5_Ni_0.5_O_4_/Li_2_MnO_3_–LiMn_0.4_Ni_0.4_Co_0.2_O_2_ composite components delivered a high capacity of 293 mAh g^−1^ at 0.1 C, showing strong capacity retention of 89.5% after 200 cycles at 1 C, and exhibited a high capacity of 202 mAh g^−1^ even at 5 C. To date, there have been few reports on the combined merits of layered and spinel materials to render a superior material in terms of kinetics, energetics, and stability [41,42,43]. Yin et al. [44] synthesized the high–voltage cathode of Li_1.2_Mn_0.54_Ni_0.13_Co_0.13_O_2_ spheres coated with thin LiNi_0.5_Mn_1.5_O_4_ layers via a simple coprecipitation method. This coated composite cathode recorded a superior rate capacity and stable cycle performance in the 2.0–4.8 V range. The Li_1.2_Mn_0.54_Ni_0.13_Co_0.13_O_2_/LiNi_0.5_Mn_1.5_O_4_ microsphere cathodes delivered a high capacity of 249.5 and 96 mAh g^−1^ at the rate of 0.2 C and 5 C, respectively. In addition, LiMn_2_O_4_/LiNi_0.5_Mn_1.5_O_4_ hollow microspheres have been developed by Liu et al. [45], with a facile solution–phase coating and subsequent solid–phase lithiation route in an atmosphere of air. Their double–shell LiMn_2_O_4_/LiNi_0.5_Mn_1.5_O_4_ hollow microspheres showed a high specific capacity of 120 mAh g^−^^1^ at a 1 C rate and an excellent rate capability (90 mAh g^−^^1^ at 10 C) over the 3.5–5 V range compared to Li/Li+, with a retention of 95% over 500 cycles.

In this experiment, a nanolayer of spinel LiCoMnO_4_ (LCMO) was used as a coating because of its voltage stability window (5.3 V vs Li/Li^+^) [46,47,48]. In this composition, lithium–rich oxide contributes to high energy density, while the spinel nanolayer of LCMO provides structural stability to maintain the surface of the layered lithium–rich cathode during high–voltage cycling. Most importantly, this high Li^+^ conductive nanolayer can rapidly transport Li^+^ between the electrolyte and the layered lithium–rich cathode. Owing to the high capacitance advantage of the lithium–rich cathode combined with the structural stability of the high–voltage spinel LCMO nanolayer, the electrochemical performance of lithium–rich cathode materials is expected to improve.

## 2. Materials and Methods

### 2.1. Cathode Preparation

The synthesis of Li_7/6_Mn_1/2_Ni_1/6_Co_1/6_O_2_ was carried out via the coprecipitation of the precursor, Mn_1/2_Ni_1/6_Co_1/6_(CO_3_)_5/6_. A 2 M solution of CoSO_4_·7H_2_O, NiSO_4_·7H_2_O, and MnSO_4_·H_2_O was dropped into a continuous stirred–tank reactor. A 2 M Na_2_CO_3_ solution and an appropriate volume of NH_4_OH solution as the chelating agent were then separately pumped into the reactor. The solutions were maintained at 60 °C, with a stirring speed of 1200 rpm and a pH of 8. After the reaction, the Mn_1/2_Ni_1/6_Co_1/6_(CO_3_)_5/6_ precursor was filtered and air–dried.

The Mn_1/2_Ni_1/6_Co_1/6_(CO_3_)_5/6_ precursor was added to the batch reactor as a core, and 1%, 2.5%, and 5% concentrations of CoSO_4_·7H_2_O and MnSO_4_·H_2_O solutions at a molar ratio of Co^2+^:Mn^2+^ = 1:1 were pumped in. A stoichiometric volume of Na_2_CO_3_ and an appropriate volume of NH_4_OH were also added. After stirring for 12 h, the precursors of Mn_1/2_Ni_1/6_Co_1/6_(CO_3_)_5/6_ coated with 0%, 1%, 2.5%, and 5% CoMn(CO_3_)_2_ were obtained. After drying, the coated precursors with 5% excess LiOH were calcined for 14 h at 900 °C to obtain Li_7/6_Mn_1/2_Ni_1/6_Co_1/6_O_2_ cathodes coated with 0% (LR–Bare), 1% (LR–Sp1), 2.5% (LR–Sp2.5), and 5% (LR–Sp5) LiCoMnO_4_, respectively.

### 2.2. Characterization

Powder X–ray diffraction (XRD) measurements of the synthesized samples were performed using an X–ray diffractometer (D8A25, Bruker, USA) operated at 40 keV and 40 mA, using Cu Kα radiation. Microstructural investigation of the powders was performed via a scanning electron microscopy (SEM, JSM–7500F, JEOL, Japan) and a high–resolution transmission electron microscope (HR–TEM, JEM–2100Plus JEOL, Japan). The particle sizes of the powders were measured using a static light–scattering analyzer (Mastersizer 2000, Malvern Instruments Ltd., UK), resulting in a measurement range of 0.04–2000 μm. The density of the cathode was measured using gas pycnometry (AccuPyc II, ATA scientific pty ltd, Australia), a non–destructive technique that employs gas displacement, often helium, to measure the volume of a material. The local structure of the sample was studied by Raman spectroscopy (MRI) using a laser with a wavelength of 532 nm and a power of 100 mW. Chemical state and composition analyses were then performed by X–ray photoelectron spectroscopy (XPS) using a Physical Electronics instrument (PHI 5000 VersaProbe Ⅲ, ULVAC–PHI. Inc., Japan) with an Al X–ray source.

### 2.3. Electrochemical Measurements

The electrochemical properties were examined using a coin cell (CR–2032) with Li foil as the anode. Cathode electrodes were prepared by mixing 80 wt% as–prepared cathodes, 10 wt% acetylene carbon black, and 10 wt% polyvinylidene difluoride (PVDF) in an n–methyl–2–pyrrolidone solution. Next, the slurries were coated onto aluminum foil. The thickness of the loaded active materials was 31.6 ± 4.2 μm, whilst the mass loading was 7.9 ± 0.2 mg cm^−2^. An organic solution of 1 M LiPF_6_ in ethylene carbonate (EC) and dimethyl carbonate (DMC) with a 1:2 volume ratio was used as the electrolyte. The charge–discharge profiles were measured at a current density of 20 mA g^−1^ (0.1 C) in the 2–4.8 V range for the formation process and then continually tested from a current density of 20 mA g^−1^ (0.1 C) to 1000 mA g^−1^ (5 C) within the 2–4.6 V range. After one formation cycle, the cycle life was determined by charging at 0.2 C and subsequent discharging at 0.5 C from 2 to 4.6 V. A second cycle life test was conducted at a high voltage of 4.9 V, with the same current density. The electrochemical impedance spectroscopy (EIS) test of the cells in reference to the Li/Li^+^ couple was measured after 25 charge–discharge cycles at 2–4.6 V and then again at a charged state of 4.2 V, within a frequency range of 0.01–100,000 Hz with a perturbation amplitude of 10 mV.

## 3. Results

### 3.1. Morphological Characterization

The surfaces and morphologies of the precursors were examined using electron microscopy, as shown in Figure 1. For the LR–Bare precursors, the nucleating cores of MCO_3_ (M = Mn, Ni, Co) grew to form spherical primary particles with a diameter of 30 nm, as shown in Figure 1b. During the coprecipitation process, the primary particles aggregated into micro secondary particles with a spherical shape, as shown in Figure 1a, having the most stable thermodynamic state and minimal surface energy [49]. The primary particles of LR–Sp1 precursors with 1% nanolayer coverage had a size of 50–70 nm. When the coating concentration increased to 2.5%, the surface of the precursors was covered with dense square grains and a few flake particles. After a further increase to 5% coating, the flake particles on the surface of the precursors became larger.

The precursors were calcined with the required amount of LiOH at 900 °C to form LR–Bare, LR–Sp1, LR–Sp2.5, and LR–Sp5 cathodes, and their morphologies are shown in Figure 2. The spherical primary particles of the LR–Bare cathode grew to 100 nm, whilst the primary particles of LR–Sp1, LR–Sp2.5, and LR–Sp5 also became spherical, with size growing alongside the increasing concentration, to eventually reach a diameter of approximately 200–300 nm. The secondary particle morphologies of all cathodes presented microsphere aggregation, and their mass median diameters (D_50_) and true densities increased with the coating concentration. The mass median diameters (D_50_) of secondary particles for LR–Bare, LR–Sp1, LR–Sp2.5, and LR–Sp5 cathodes were 7.2, 7.8, 8.6, and 8.9 μm, respectively, as shown in Figure 3. Meanwhile, the true densities of these cathodes were 4.02, 4.12, 4.18, and 4.28 g cm^−1^, respectively.

### 3.2. Structural Characterization

The structural characterization XRD patterns for all samples are plotted in Figure 4. The main peaks in the diffraction patterns of LR–Bare, LR–Sp1, LR–Sp2.5, and LR–Sp5 could be identified as that for hexagonal α–NaFeO_2_, with the space group of R$\bar{3}$m. On the other hand, the low–intensity superlattice peaks in the 2θ range of 20–25° could be attributed to the Li_2_MnO_3_–like component, with a space group of C2/m and the ordering of Li, Ni, and Mn atoms in the transition metal layers [50,51]. In the coated samples, the spinel structure of LiCoMnO₄ (LCMO) was not identified because of the cubic close–packing structures. This presented similar peaks with the XRD patterns of layered and spinel lithium metal oxides, making it difficult to distinguish the layered and spinel components from coated samples with low spinel content.

To further discern these spinel and layered structures, Raman spectroscopy was used to determine the short–range local structures of the materials. Figure 5 displays the Raman spectra of the four samples. The Raman bands located at 590, 471, and 425 cm^−1^ were observed in all samples, presenting typical characteristics of layered lithium–rich oxides, indicating that the layered structure was maintained after the deposition of the spinel nanolayer coating. There were two vibrational modes for the lithium–rich oxides with hexagonal R$\bar{3}$m symmetry: E_g_ with symmetrical deformation and A_1g_ with the symmetrical stretching of M–O (M = Mn, Ni, and Co) [52]. The weak band at 425 cm^−1^ was the characteristic vibration of Li_2_MnO_3_, which has monoclinic C2/m symmetry and a lower local symmetry than R$\bar{3}$m [40]. Meanwhile, two additional shoulder bands at around 635 and 551 cm^−1^ and another peak at 481 cm^−1^ in LR–Sp1, LR–Sp2.5, and LR–Sp5, as indicated by the blue arrows, were typical peaks of the spinel–phase with Fd3m symmetry [53,54]. Overall, the Raman spectra presented mixed characteristics of the spinel and layered phases for the coated samples. 

To investigate the surface chemical states of the samples, X–ray photoelectron spectroscopy (XPS) measurements were obtained, as shown in Figure 6. The Ni 2p_3/2_ peaks at 854.5 and 860.9 eV indicated the coexistence of Ni^2+^ and Ni^3+^, as shown in Figure 6a [55]. In addition, the binding energy of 779.9 eV for the Co 2p_3/2_ peak was assigned to the Co^3+^ species, shown in Figure 6b [56]. Figure 6c shows that Mn p_3/2_ and Mn 2p_1/2_ had binding energies of 641.9 and 653.2 eV, respectively, which are typical values for Mn^4+^ [57,58]. The Mn, Ni, and Co XPS peaks for LR–Bare and LR−Sp2.5 presented the same binding energies. Figure 6d compares the O 1s peaks; there were two pronounced peaks at 528.9 and 530.6 eV, which corresponded to lattice oxygen atoms (O–M–O) and oxygen vacancies, respectively. The O 1s peak areas of LR–Sp2.5, especially the peak of oxygen vacancies, were smaller than those for LR–Bare, which indicated that the LCMO spinel nanolayer was successfully constructed on the surface of the layered lithium–rich oxides, thus exhibiting good structural compatibility.

### 3.3. Electrochemical Characterization

The charge–discharge rate performances of the samples at 0.1 C over 2.0–4.8 V during the formation process and at 0.1, 1, and 5 C between 2.0 and 4.6 V are presented in Figure 7. LR–Bare had the lowest rate capability, alongside a dramatic capacity drop with an increase in C–rate; this was because of the fast Li^+^ insertion and extraction, which damaged the fragile surface of the layered structure at a high rate. The C–rate results for LR–Sp1 were similar to those for LR–Bare because of the low concentration of the spinel nanolayer. LR–Sp2.5 and LR–Sp5, especially the former, exhibited superior discharge capacities and C–rate performances compared to LR–Bare. These results indicate that the Li–rich layered oxide with LCMO spinel nanolayer provided the combined advantages of high capacity from the Li–rich layered oxide and high rate capability from the spinel nanolayer. The discharge capacities of LR–Bare were 271, 254, 193, and 144 mAh g^−1^ at 0.1 C and 2.0–4.8 V during formation and at rates of 0.1, 1, and 5 C across 2.0−4.6 V, respectively. Under the same conditions, LR–Sp2.5 exhibited excellent discharge capacities of 300, 275, 220, and 166 mAh g^−1^, respectively. Thus, it is believed that an optimal concentration of the LCMO spinel nanolayer has good structural compatibility with Li–rich layered oxides and can provide relatively high working C–rates owing to 3D interstitial spaces [59]; this accelerates lithium–ion insertion and extraction kinetics [60,61]. 

After formation at 2.0–4.8 V, the cycle performance of the samples was tested for 100 cycles at 0.2 C/0.5 C charge–discharge between 2.0 and 4.6 V, with these results shown in Figure 8. When cycling for 60 cycles, the capacities of all samples were stable at approximately 200 mAh g^−1^; however, after 60 cycles, the capacity of LR–Bare began to decrease rapidly. After 100 cycles, the discharge capacity of LR–Bare was 150.5 mAh g^−1^. In contrast, remarkably improved cycling stability was observed in the samples with the LCMO spinel nanolayers. Even after 100 cycles, the discharge capacities of LR–Sp1, LR–Sp2.5, and LR–Sp5 remained at 183.7, 193.6, and 192.1 mAh g^−1^, respectively. This is because the spinel nanolayer prevented phase transition on the Li–rich layered oxide surface during cycling, thus improving the cycle performance. Therefore, it prevented the generation of stress induced by the anisotropic volume change of the primary crystallites in secondary particles, further preventing the generation of cracks [58]. Overall, samples with spinel nanolayers displayed better cycle performance [62]. 

To explain the electrochemical properties of LR–Bare and LR–Sp2.5, electrochemical impedance spectroscopy (EIS) was operated after 25 charge–discharge cycles at 2.0–4.6 V and then in a charged state at 4.2 V. The impedance spectra shown in Figure 9 contained the bulk resistance of the electrolyte (R_e_), the resistance at the contact of the composite cathode (R_SEI_), the charge transfer resistance at the interface of the cathode–electrolyte (R_ct_), and a line inclined at a constant angle to the real axis in the low–frequency range corresponding to Li^+^ diffusion in the materials [5,55,58]. The R_ct_ and lithium ion diffusion coefficient (D_Li_^+^) were 23.1 Ω and 2.96 × 10^−12^ cm^2^ s^−1^ for LR–Bare and 20.0 Ω and 3.21 × 10^−12^ cm^2^ s^−1^ for LR–Sp2.5, respectively. This shows that the sample with a spinel nanolayer had the fastest lithium–ion insertion and extraction kinetics at the electrolyte–electrolyte interface.

Transmission electron microscopy (TEM) was further utilized to study the detailed morphological structures of LR–Bare and LR–Sp2.5 samples. LR–Bare is an agglomerated secondary particle with nanosized primary grains, as shown in Figure 10a. In Figure 10b, LR–Bare presents a smooth edge line without any other layer on the surface of the grain. During long–term cycling, partial cation ordering in the layered structure of LR–Bare was locally rearranged into an inactive phase that contained transition metal ions in both the lithium and transition metal layers [63]. The phase transformation accompanied the lattice strain in the grains; consequently, nanosized domains were identified in LR–Bare grains, which developed into a polycrystalline structure after 100 cycles, as shown in Figure 10c,d. After modification with the LCMO spinel nanolayer, LR–Sp2.5 was covered by a thin film with a width of 5–8 nm, as shown in Figure 10e. Throughout 100 cycles, LR–Sp2.5 exhibited polycrystalline and intact grains, as shown in Figure 10f,g. According to Gu [64] and Xu et al. [12], the migration of transmission metal ions to Li layers leads to a layered–to–inactive spinel phase transformation, which only occurs at the surface of the particles in the beginning. During long–term cycling, the inactive spinel phase transformation extends to the inner parts of the particles. In our study, after 100 cycles, LR–Sp2.5 presented mosaic grains coated with a spinel nanolayer (Figure 10f) and undamaged particles without a coating (Figure 10g). Therefore, we presume the former was likely located on the outer surface, and the latter was present in the center of the microsphere aggregates of the LR–Sp2.5 particle. This proves that the LCMO spinel nanolayer could effectively stabilize the structure and relieve the particle damage caused by phase transformation, thereby protecting the inner particles, and allowing them to function normally. 

To further understand the utility of the LCMO spinel nanolayer under high voltage on the Li–rich layered oxide, the range of the charge–discharge voltage was raised to 2.0–4.9 V. The cycle performances of the samples at 0.2 C and 0.5 C and charge–discharge at 4.9 V are shown in Figure 11a. In this case, the capacity of LR–Bare decayed very quickly with an increasing number of cycles owing to the high voltage, and the battery completely failed after 42 cycles. On the contrary, the samples with the LCMO spinel nanolayers showed better cycling performance, with LR–Sp2.5 having the best and most stable capacity retention. Figure 11b–e shows the discharge profiles of LR–Bare, LR–Sp1, LR–Sp2.5, and LR–Sp5 for cycles 10, 20, 30, and 40, respectively. In Figure 11b, LR–Bare delivered a capacity of 228.5 mAh g^−1^ at the 10th cycle, but a capacity of only 28.1 mAh g^−1^ remained after 40 cycles, alongside significant voltage fading. For LR–Sp1 and LR–Sp5 in Figure 11c,e, the capacity was approximately 230 mAh g^−1^ at the 10th cycle. However, capacity and voltage had rapidly decayed by the 20th cycle before stabilizing during later cycles. However, LR–Sp1 completely failed after 45 cycles because of the low concentration of the LCMO spinel nanolayer. In Figure 11d, LR–Sp2.5 demonstrated significant improvements in both specific capacity and voltage maintenance with increasing cycles at 2.0–4.9 V. LCMO is a high–voltage cathode material with an operating voltage of approximately 5.3 V [48]; therefore, its electrochemical performance is stable at 4.9 V. When LCMO is used as a coating layer, it can yield good structural compatibility to stabilize the surface of the Li–rich layered oxide and avoid the generation of oxygen vacancies, which is induced by the phase transformation from the layered structure to the inactive spinel structure at the surface. In addition, LCMO can suppress the side reactions caused by the electrolyte at high voltages. These features enhance the diffusion kinetics of Li ions, improve cycling performance, and alleviate voltage decay.

The differential capacity vs. voltage (dQ/dV) curves provide a clear perspective on the electrochemical reactions that occur during cycling. Figure 12 shows the dQ/dV curves of LR–Bare and LR–Sp2.5 cathodes cycled at 0.2 C and 0.5 C across 2.0–4.9 V. There were three redox reactions corresponding to Mn, Ni, Co, and O. For LR–Bare, two anodic peaks of Mn^3+/4+^ were observed; the first one, at 3.1 V, was induced by Mn in an inactive spinel structure and the other one, at 3.3 V, was related to Mn in the Li–rich layered oxides [65,66]. A prominent oxygen–activating peak also appeared at approximately 4.5 V, which was related to the layered–to–inactive spinel phase transformation of Li–rich layered oxides [67,68,69,70], as shown in Figure 12a. In Figure 12b, the divided peak of Mn was not observed, and the intensity of the O^2−^ anodic peak for LR–Sp2.5 was much lower than those of LR–Bare. In addition, during the cycling, LR–Sp2.5 presented greater overlap among the cathodic peaks of Mn compared with LR–Bare. This indicates that LR–Sp2.5 exhibited inconspicuous polarization at a higher cutoff voltage of 4.9 V than LR–Bare. After cycling at high voltage, the LCMO spinel nanolayer maintained the structural stability of the lithium–rich oxide surface, thereby suppressing phase transformation and crystallographic distortions.

## 4. Conclusions

Lithium–rich oxides, Li_7/6_Mn_1/2_Ni_1/6_Co_1/6_O_2_, were successfully modified with a LiCoMnO_4_ (LCMO) spinel nanolayer on the surface using a coprecipitation method. The LCMO spinel nanolayer effectively prevented side reactions between the lithium–rich oxide and organic electrolytes, thereby promoting ion and charge transfer on the surface of the lithium–rich oxides. Meanwhile, the LCMO nanolayer could effectively restrain the migration of transmission metal ions and the generation of oxygen vacancies induced by the phase transformation from the layered structure to the inactive spinel structure at the surface of lithium–rich oxides. As a result, the obtained LR–Sp2.5 showed excellent cycle and rate performances during cycling at normal (4.6 V) and high (4.9 V) voltages.

## Figures and Tables

**Figure 1 nanomaterials-12-03425-f001:**
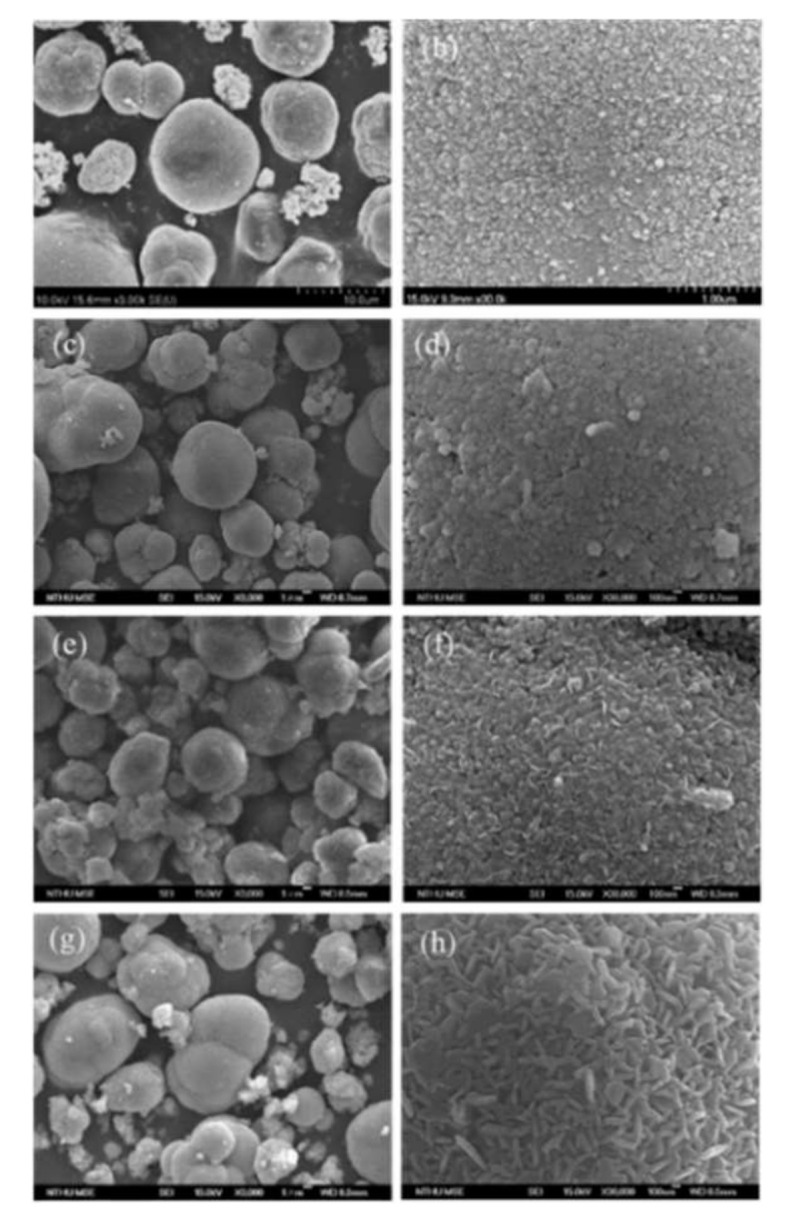
SEM images of (**a**,**b**) LR–Bare, (**c**,**d**) LR–Sp1, (**e**,**f**) LR–Sp2.5, and(**g**,**h**) LR–Sp5 precursors.

**Figure 2 nanomaterials-12-03425-f002:**
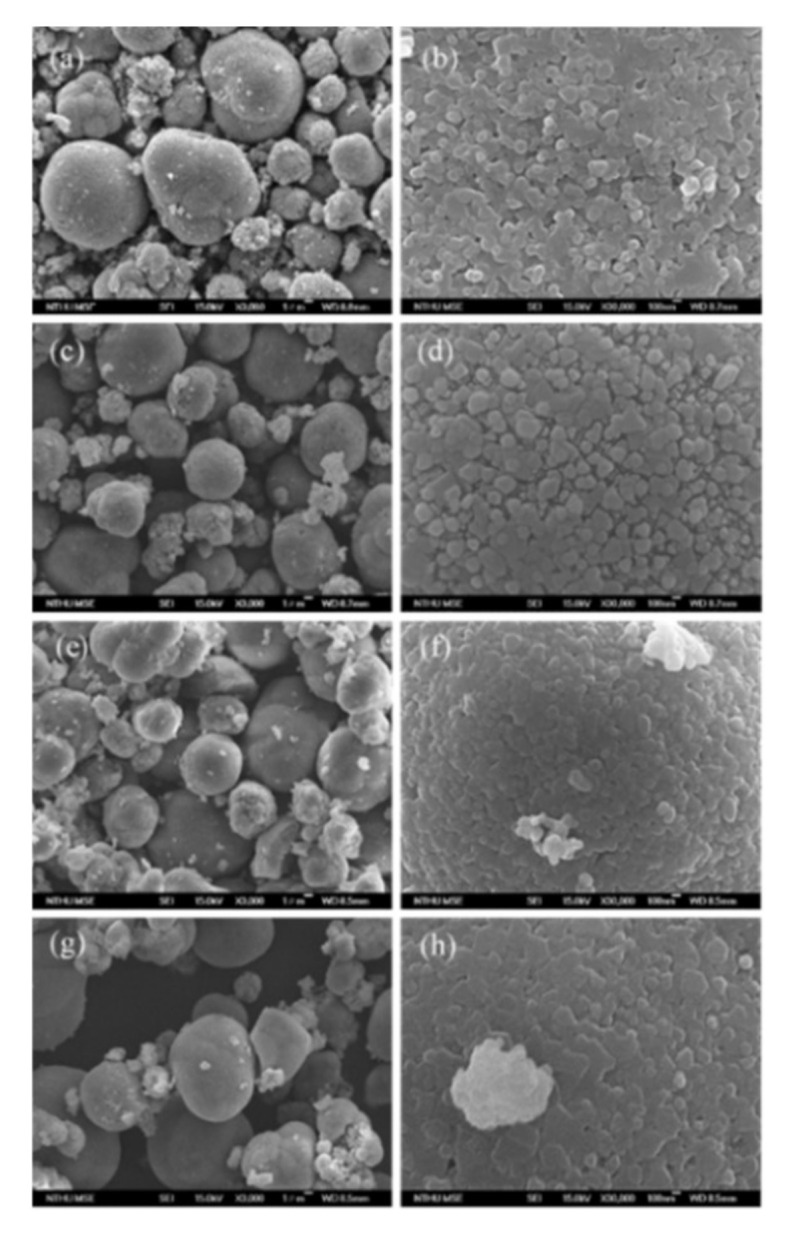
SEM images of (**a**,**b**) LR–Bare, (**c**,**d**) LR–Sp1,(**e**,**f**) LR–Sp2.5, and (**g**,**h**) LR–Sp2.5 cathodes.

**Figure 3 nanomaterials-12-03425-f003:**
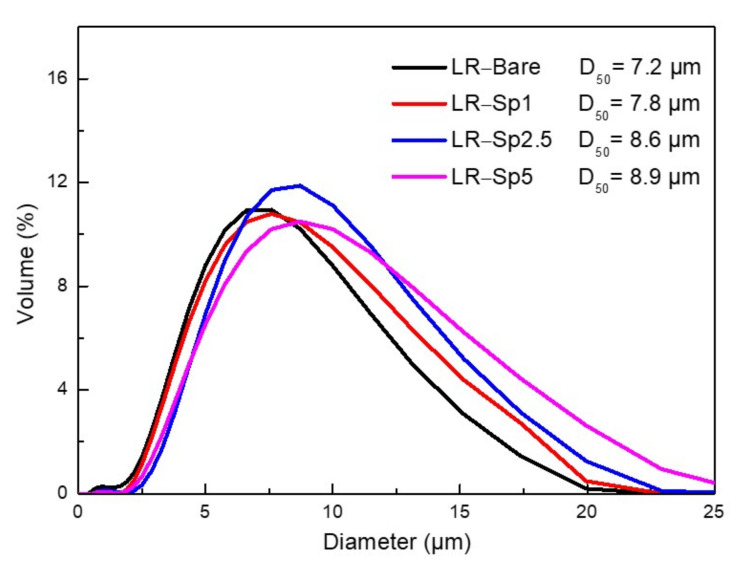
Size distribution of each cathode.

**Figure 4 nanomaterials-12-03425-f004:**
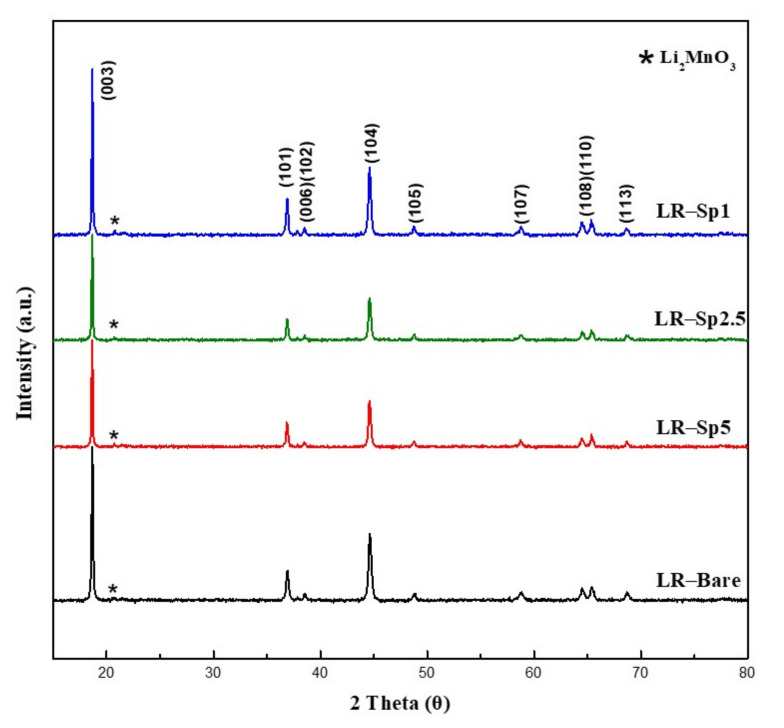
X–ray diffraction (XRD) patterns of the LR–Bare, LR–Sp1, LR–Sp2.5, and LR–Sp5 cathodes.

**Figure 5 nanomaterials-12-03425-f005:**
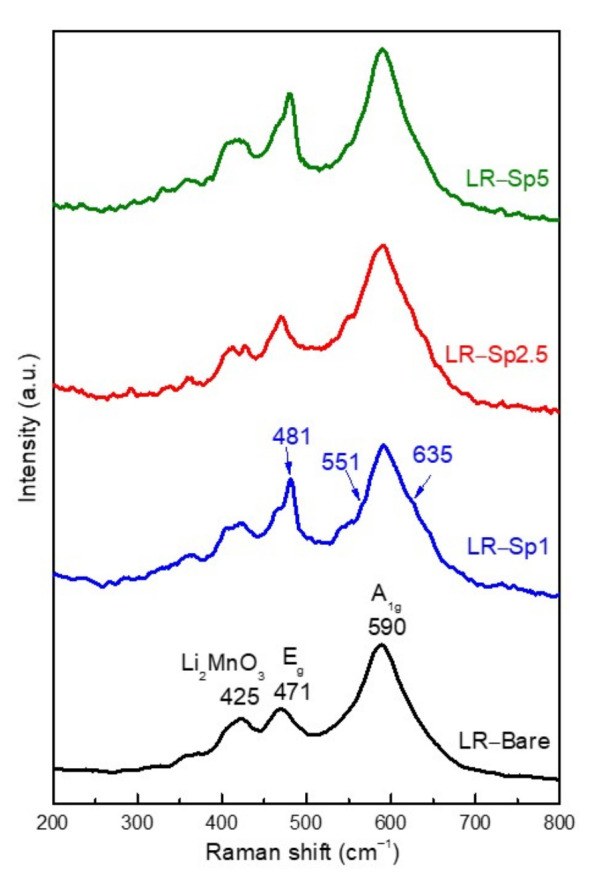
Raman spectra of the LR-Bare, LR-Sp1, LR-Sp2.5, and LR-Sp5 cathodes.

**Figure 6 nanomaterials-12-03425-f006:**
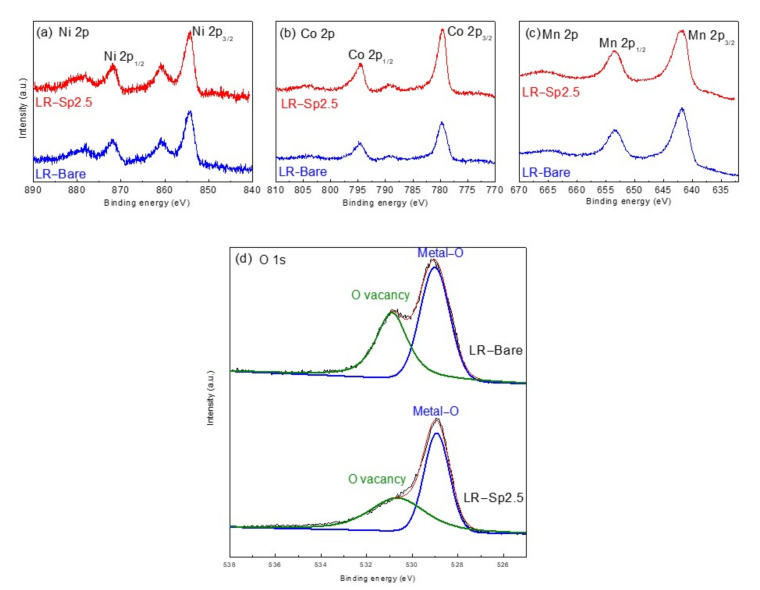
XPS spectra of (**a**) Ni 2p, (**b**) Co 2p, (**c**) Mn 2p, and (**d**) O 1s for LR–Bare and LR–Sp2.5 cathodes.

**Figure 7 nanomaterials-12-03425-f007:**
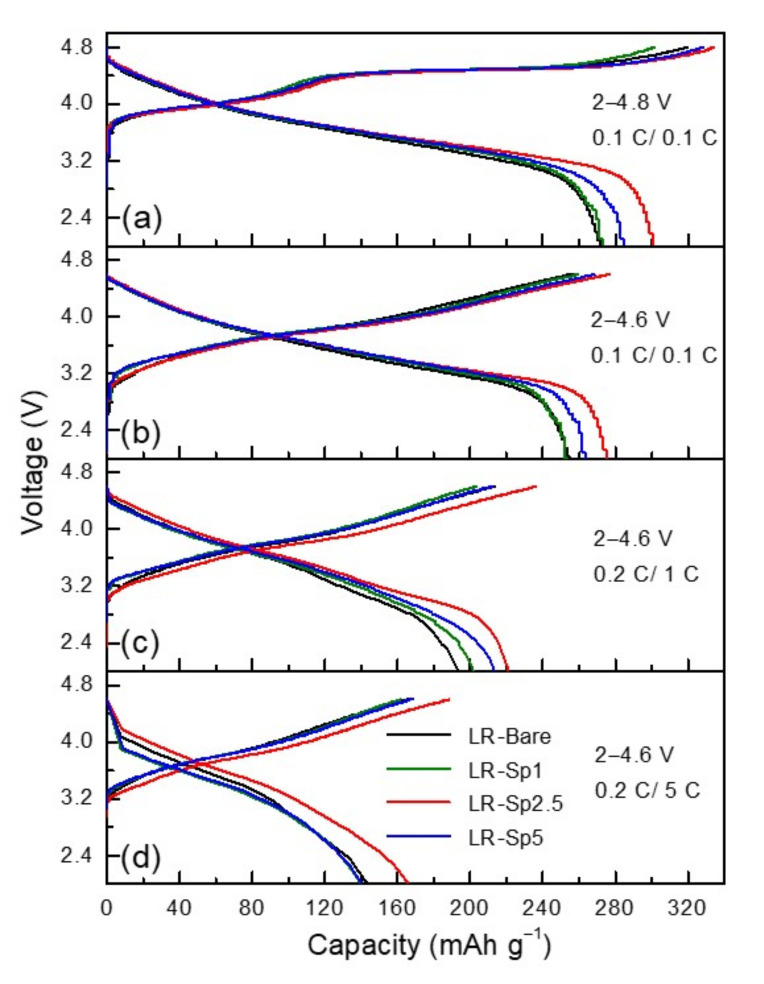
Charge and discharge performance at (**a**) 0.1 C/0.1 C between 2–4.8 V, (**b**) 0.1 C/0.1 C, (**c**) 0.2 C/1 C and (**d**) 0.2 C/5 C between 2–4.6 V for the LR–Bare, LR–Sp1, LR–Sp2.5, and LR–Sp5 cathodes.

**Figure 8 nanomaterials-12-03425-f008:**
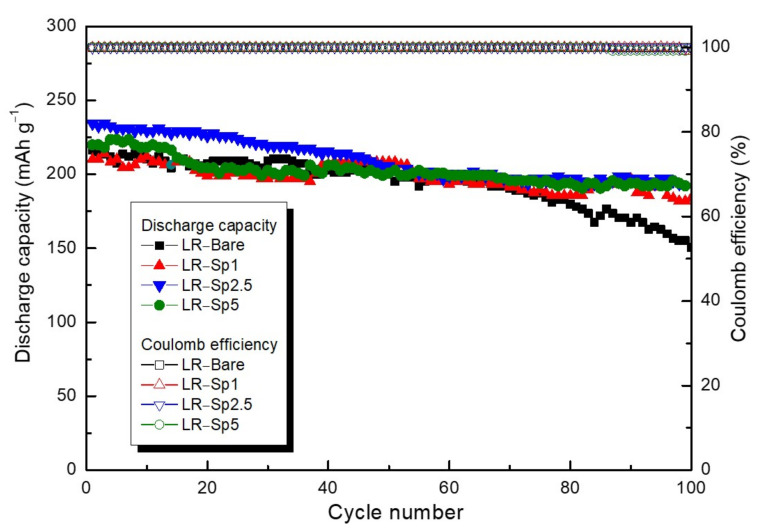
Cycle life performance of LR–Bare, LR–Sp1, LR–Sp2.5, and LR–Sp5 cathodes at 2–4.6 V.

**Figure 9 nanomaterials-12-03425-f009:**
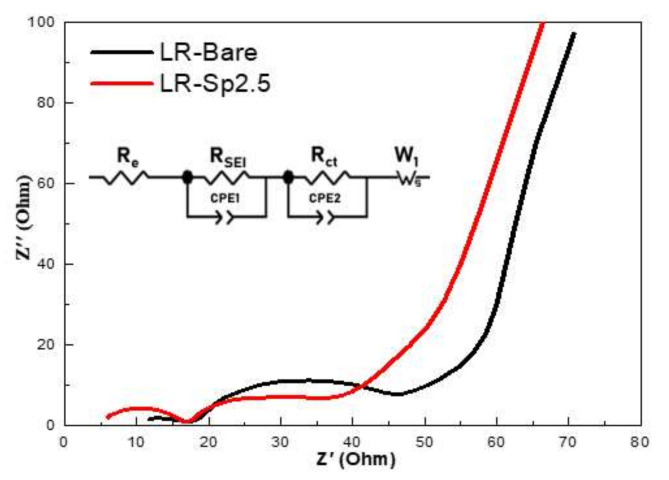
Electrochemical impedance spectra of LR–Bare and LR-Sp2.5 after 25 charge–discharge cycles.

**Figure 10 nanomaterials-12-03425-f010:**
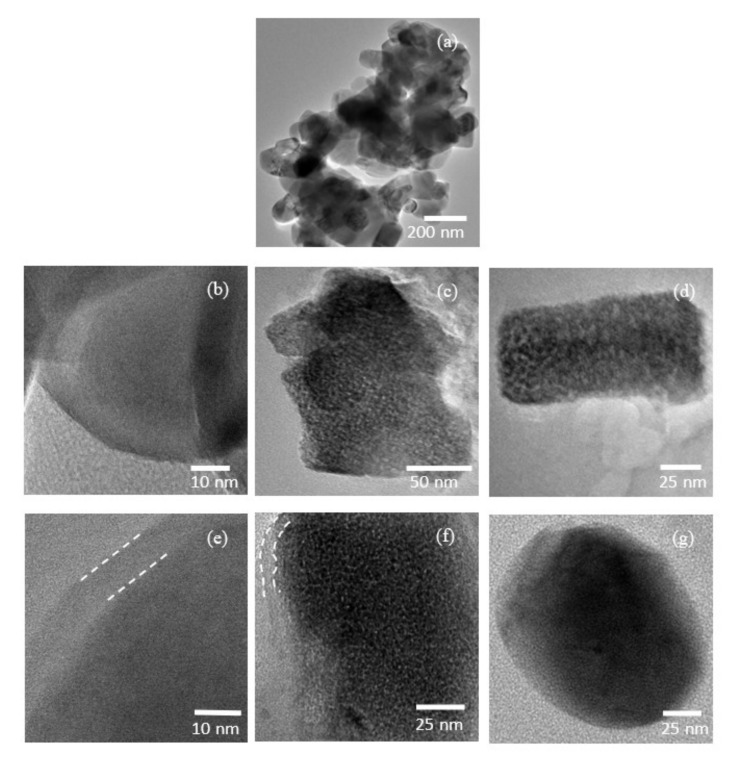
Transmission electron microscopy (TEM) images of (**a**,**b**) LR–Bare before cycle and (**c**,**d**) after 100 cycles; and (**e**) LR–Sp2.5 before cycle and (**f**,**g**) after 100 cycles.

**Figure 11 nanomaterials-12-03425-f011:**
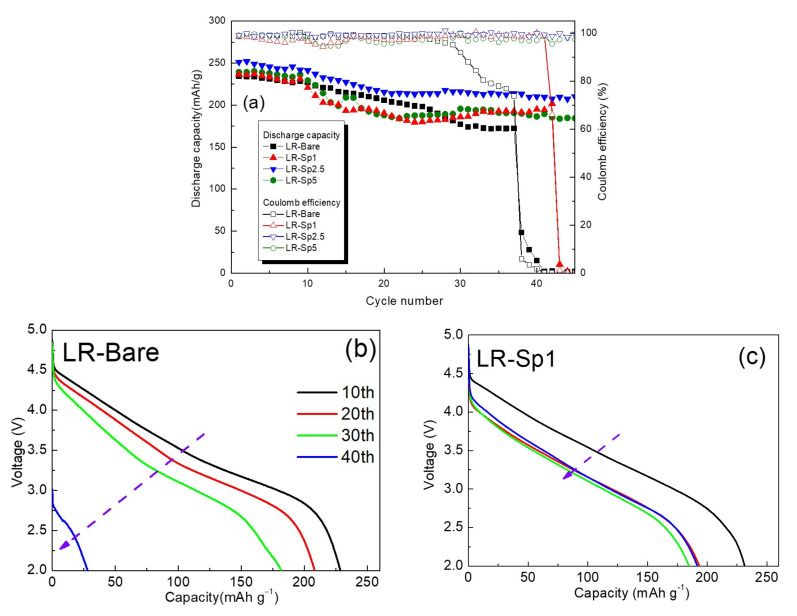
(**a**) Cycle life performance and discharge profiles for the 10th, 20th, 30th, and 40th cycles for (**b**) LR–Bare, (**c**) LR–Sp1, (**d**) LR–Sp2.5, and (**e**) LR–Sp5 cathodes at 2.0–4.9 V.

**Figure 12 nanomaterials-12-03425-f012:**
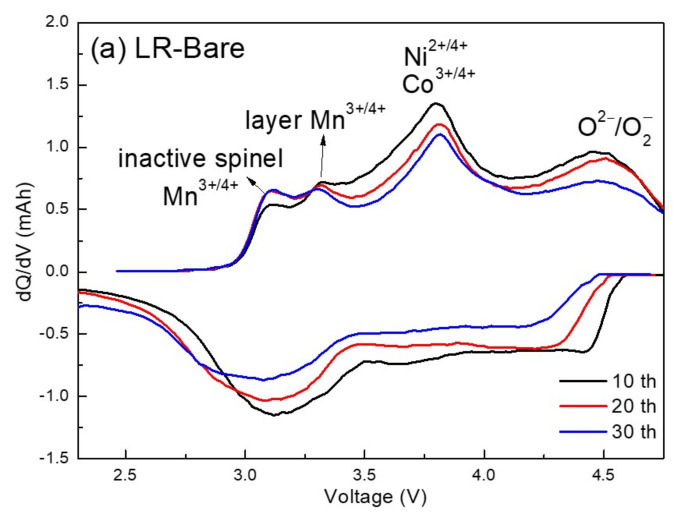
Differential capacity curves of (**a**) LR–Bare and (**b**) LR–Sp2.5 cathodes for the 10th, 20th, and 30th cycles.

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
