# Peer review of "Stabilizing Li-Rich Layered Cathode Materials Using a LiCoMnO4 Spinel Nanolayer for Li-Ion Batteries"

_nanomaterials, 2022, doi:10.3390/nano12193425_

Round 1
Reviewer 1 Report
This manuscript offers a concise yet compelling report on the stabilizing Li-rich layered cathode materials by LiCoMnO4 spinel nanolayer for Li-ion batteries. The emphasis of the study falls on the achieved by the nanolayer spinel coating phase transformation that reliably enhances the electrochemical performance of the cathode. The discussion provided is quite adequate for the present ambitious purpose as well as the focused and convincing characterization effort. The nice detailed and at the same time comparative context of the XRD, SEM, Raman results, among others, and all the subsequently provided explanations throughout the study contributes to a reliable scrutinization and prompting good understanding of possibly for repeatable realization of cathode materials enhanced by LiCoMnO4 spinel nanolayer as well as paving the way to future applications with potentially high practical impact for Li-ion batteries.
From practical point of view, the reported results thus bring new knowledge and certainly represent an original contribution in the present context.
The authors chose an adequate structure of the manuscript – an excellent point of departure for such a study. Finally, the authors provided a balanced realistic and nicely illustrated presentation of their results and corresponding analysis that is of much scientific and practical interest and adds new knowledge to the field.
In my opinion, the fine detailing in the present work, the insightful and balanced discussion of the results, as well as the excellent, intuitively perceived figures, permit wide circle of readers to utilize the manuscript as a guidance for their potential future work in the same or in a similar research field. Consequently, this manuscript presents an efficient and beneficial basis for promoting and solving next step challenges in this field.
The manuscript also benefits from a clear motivation, and it is an easy and informative read.
The present manuscript is a significant contribution, this work once published would be quite useful as well as instructive and suggestive in terms of further studies and to a wider readership.
There are some minor issues with this already excellent manuscript that will need to be addressed before becoming suitable for publication, i.e., it can be considered for publication after a minor revision:
1: In the introduction, the authors partly miss that previously a very wide range of theoretical/simulation approaches/including by using first-principles calculation tools have already been used to study nanostructured coating phase transformation/stabilization in different context of inherently nanostructured coatings and heterostructured materials. Nowadays, examples in which such first-principle works help understanding the mentioned synergies and structural issues and guide experimental work include Journal of Physics: Condensed Matter 27 (2015) 485306, ACS applied materials & interfaces 10 (2018) 16238-16243; Such works should be referred to.
2: The authors should elaborate and be more specific when they comment on (good) thermal stability. Are there any direct limitations of its thermal stability in the present material system?
3: It would be helpful and valuable to the general readership if bonding nature is commented in more quantitative details and if it is placed in the larger context of bonding in similar lithium-rich cathode materials.
4: Spell-check and stylistic revision of the paper are still necessary. Some, long sentences, misspellings, etc., still are noticeable throughout the text.
Reviewer 2 Report
Recommendation: minor revision.
Comments: In this work, the authors prepared LiCoMnO4 modified Li7/6Mn1/2Ni1/6Co1/6O2 lithium-rich cathodes that were synthesized via a co-precipitation method. The structure and physicochemical properties of as-prepared materials were studied in detail with suitable techniques and reasonably explained. The electrochemical performance of the LiCoMnO4 modified Li7/6Mn1/2Ni1/6Co1/6O2 cathodes for LIBs is investigated. The experiment data relevant to LiCoMnO4 modified Li7/6Mn1/2Ni1/6Co1/6O2 cathode offered in this manuscript are sufficient to support the conclusion. So, I recommend that this manuscript can be accepted for publication in Nanomaterials after minor revision.
1. The introduction of this paper needs to make a strong argument about the impact and novelty of the work further. So, the introduction should enrich some related cathodes in this section (such as Nanomaterials, 2022, 12, 1888; Chem. Eur. J., 2014, 20, 824-830; RSC Adv., 2015, 5, 84673).
2. The thickness of the loaded active materials can be evaluated. The mass loading of active materials is better offered.
3. The Figures should be refined or revised. Some Figures are overlapped, such as Figure 6 and Figure 10.
4. To evaluate these Li1.0[Ni0.8Co0.1Mn0.1]O2 cathodes undergo, the HRTEM images before or after cycles are better offered.
5. The corresponding coulombic efficiency of Figure 8c is better added to the manuscript.
6. The Li+ diffusion coefficient of the Li7/6Mn1/2Ni1/6Co1/6O2 and LiCoMnO4 modified Li7/6Mn1/2Ni1/6Co1/6O2 is better calculated.
7. The authors better compare the electrochemical performance of the LiCoMnO4 modified Li7/6Mn1/2Ni1/6Co1/6O2 with reported Ni-rich layered cathode materials.
8. Some issues and writing mistakes exist in the manuscript. The authors should carefully check and correct them. Such as “at a current density of 20 mAh g-1 (0.1 C)” on line 118 should be“at a current density of 20 mA g-1 (0.1 C)”
